# Early Transpyloric Tube Feeding in Preventing Adverse Respiratory Events in Extremely Low Birth Weight Infants

**DOI:** 10.3390/biomedicines12122799

**Published:** 2024-12-10

**Authors:** Shinya Tanaka, Fumihiko Namba, Ken Nagaya, Naohiro Yonemoto, Shinya Hirano, Itaru Yanagihara, Hiroyuki Kitajima, Masanori Fujimura

**Affiliations:** 1Department of Neonatology, Osaka Women’s and Children’s Hospital, Izumi 594-1101, Japan; tanakababykids@gmail.com (S.T.); nagaya5p@asahikawa-med.ac.jp (K.N.); shirano@wch.opho.jp (S.H.); kitajima@wch.opho.jp (H.K.); mfuji0415@gmail.com (M.F.); 2Department of Pediatrics, Saitama Medical Center, Saitama Medical University, Kawagoe 350-8550, Japan; 3Division of Neonatology, The Center of Maternity and Infant Care, Asahikawa Medical University Hospital, Asahikawa 078-8510, Japan; 4Department of Biostatistics, Faculty of Medicine, University of Toyama, Toyama 930-0194, Japan; nyonemoto@gmail.com; 5Department of Clinical Trials Office, Osaka Women’s and Children’s Hospital, Izumi 594-1101, Japan; 6Department of Developmental Medicine, Osaka Women’s and Children’s Hospital, Izumi 594-1101, Japan; itaruy@wch.opho.jp

**Keywords:** transpyloric tube feeding, nasogastric tube feeding, bronchopulmonary dysplasia, extremely low birth weight infant, adverse events

## Abstract

Background: It has been demonstrated that aspiration during endotracheal intubation in preterm infants with gastroesophageal reflux is a contributing factor in the worsening of lung diseases and the development of bronchopulmonary dysplasia (BPD). This study aims to compare the safety and efficacy of early transpyloric (TP) tube feeding with that of nasogastric (NG) tube feeding in relation to BPD. Methods: The study population consisted of 39 extremely low birth weight infants (ELBWIs) with mechanical ventilation and an enteral feeding volume of 50 mL/kg/day, which were randomly assigned to different groups based on the method of tube feeding. The primary outcome was the incidence of adverse events. Results: The hazard ratio for primary adverse events was significantly lower in the TP group. The TP group had a median time of 34 days (range 24–85) and the NG group 24 days (range 13–70). In general, neither group exhibited severe intestinal complications or poor growth. Conclusions: Early TP tube feeding may be a safer alternative method of NG tube feeding for intubated ELBWIs and has been shown to reduce the frequency of adverse respiratory events.

## 1. Introduction

The recent advancements in perinatal medicine have led to significant improvements in the protective care of the lungs of extremely low birth weight infants (ELBWIs). However, despite these advances, the incidence of bronchopulmonary dysplasia (BPD) remains a significant concern in this population [1,2,3]. The pathophysiological characteristics of BPD are complicated by multiple factors, including the immaturity of the lungs, antenatal inflammation, and the use of oxygen and mechanical ventilation [4]. Various modalities have been used clinically to prevent and/or treat BPD, including surfactant replacement therapy, high-frequency oscillatory ventilation, and administrations of antenatal, postnatal systemic and inhaled steroids, vitamins, and low-dose antibiotics. However, these treatments have not shown to reduce the incidence of BPD, contrary to expectation [3].

Gastroesophageal reflux (GER) is a common occurrence in preterm infants, with the potential to precipitate respiratory complications and inadequate weight gain. The etiology of GER is complex and influenced by multiple factors, including the supine position, immaturity of the lower esophageal sphincter, limited stomach capacity, delayed gastric emptying, and the presence of an NG tube in preterm infants [5,6]. In ELBWIs, it has been reported that tracheal silent aspiration and small amounts of refluxed gastric contents without clear symptoms do not significantly influence the development of respiratory failure [7]. The prevalence of tracheal silent aspiration in tracheally-intubated premature infants has been estimated to be as high as 80% [8]. Symptomatic GER is considered a risk factor for silent aspiration in intubated infants. Therefore, it can be suggested that good control of GER reduces the risk of silent aspiration [9,10,11]. Moreover, aspiration during endotracheal intubation in preterm infants with GER has been shown to contribute to the worsening of lung diseases and the development of BPD [12,13].

To prevent the occurrence of aspiration and reduce the incidence of respiratory failure, transpyloric (TP) tubes are commonly used for enteral feeding in preterm infants. A Cochrane review published in 2013 concluded that the available data do not provide evidence of any beneficial effect of TP tube feeding for preterm infants. Conversely, there is some evidence of harm, including a higher risk of gastrointestinal disturbance and mortality [14]. However, a retrospective study of 59 ELBWIs revealed that the duration of supplemental oxygen use was significantly shorter in the TP tube feeding group than in the nasogastric (NG) tube feeding group [15]. Moreover, the incidence of adverse events associated with TP tube feeding was only 8%, with bile-stained fluid regurgitation occurring without any severe complications.

These findings led us to investigate the relationship between the nutritional route of the TP tube and adverse effects, particularly those affecting respiratory functions, in intubated ELBWIs. We hypothesized that early TP tube feeding would reduce the incidence of BPD, which is one of the most serious complications. We then re-evaluated TP tube feeding in the context of lower mortality rates and a high incidence of BPD in ELBWIs.

## 2. Materials and Methods

### 2.1. Setting and Design

This study was conducted in the neonatal intensive care unit at Osaka Medical Center and Research Institute for Maternal and Child Health. Prior to the study, approval was obtained from the human research ethics committee (trial ID: UMIN000001728), and the study was conducted in accordance with the principles of the Declaration of Helsinki [16]. Before enrollment, informed parental consent was obtained. Once the feeding volume reached 50 mL/kg/day, which was identified as a point at which the risk of aspiration increased while the risk of intestinal disorders decreased [17], the patients were randomly divided into two groups using sealed, opaque envelopes. The patients were stratified by the presence of intrauterine infection/inflammation (high neonatal serum IgM levels, chorioamnionitis, and a negative gastric stable microbubble test [18]) and gestational age (three blocks: 22–23, 24–26, and 27 weeks). The patients were monitored until they reached a postmenstrual age of 36 weeks.

### 2.2. Participants

The study population comprised infants born prematurely between July 2001 and April 2003 who met the following criteria: birth weight of less than 1000 g, gestational age of less than 28 weeks, mechanical ventilation, enteral feeding volume of 50 mL/kg/day, and parental consent. Gestational age was determined by combining maternal dates with early prenatal ultrasound assessment. Infants with intestinal disorders, including necrotizing enterocolitis (NEC), and those with congenital anomalies, including congenital heart disease, chromosomal anomalies, and hydrops, were excluded from the study. Furthermore, all infants were administered a probiotic supplementation (*Bifidobacterium breve*) on a daily basis from the first day of life in order to prevent NEC [19,20].

### 2.3. Intervention

The infants were fed via an NG tube until the total volume of feeding reached 50 mL/kg/day. A commercially available TP tube (5 Fr, New Enteral Feeding Tube, Nippon Sherwood Medical Industries Ltd., Tokyo, Japan) was used in the TP group. The distance from the oral cavity to the pylorus along the greater curvature of the stomach was measured. Subsequently, the TP tube was passed through the mouth and the tip was successfully guided into the pylorus by gentle manual manipulation over the abdomen at the bedside. Subsequently, the tube was advanced until the bile-stained fluid could be aspirated or, if that proved impossible, until resistance was encountered. The infants were positioned in a semi-prone position on the right side, and the TP tube was advanced gradually via peristaltic movement. To confirm the position of the TP tube within the duodenum, abdominal X-rays were conducted. The optimal location for the tip of the TP tube was considered to be the third or fourth portion of the duodenum. In the event that an infant was unable to continue feeding in the allocated group, an infant in the NG group was promptly transitioned to TP tube feeding, or an infant in the TP group was similarly switched to NG tube feeding. Moreover, the follow-up process was continued.

A safe technique was implemented for the handling of the TP tube in infants with intestinal functional problems, including meconium diseases or severe acute infection from birth. The tube was carefully inserted into the duodenum to avoid injury to the intestinal mucosa in ELBWIs, which can be easily damaged by the tip of the tube during the process. The NG tube was also meticulously positioned within the nasal cavity and advanced into the gastric region, subsequent to the determination of the requisite length of tube necessary to achieve gastric access in the infant. Both groups were fed intermittently with only human milk for more than one hour.

### 2.4. Measurements

The baseline data set included the following variables: gestational age, birth weight, birth height, birth head circumference, sex, Apgar score, prolonged rupture of the membranes (>24 h), the administration of antenatal corticosteroids, the use of surfactants and diuretics, and the presence of chorioamnionitis of the placenta.

The primary outcome was the incidence of adverse events as follows: (1) BPD at 36 weeks postmenstrual age (Shennan’s definition [21]); (2) respiratory deterioration (as indicated by a ventilatory index (VI) of >0.1 or fraction of inspired oxygen (FSp95) of >0.6 for more than eight hours); (3) intestinal disorder, including cases of more than stage two of NEC; and (4) death.

The secondary outcome variables included growth variables, such as the days when feeding commenced and when the volume of feeding reached 50 and 100 mL/kg/day, daily feeding intake, body weight (measured weekly), head circumference (measured weekly), the rate of maximal weight loss, and the day of return to birth weight. The respiratory variables included the ventilatory index, FiO_2_, FSp95, and room air saturation (RAS, %). These data were recorded on a daily basis until the postmenstrual age of 36 weeks. Furthermore, the times of surfactant replacement therapy, durations of supplemental oxygen, mechanical ventilation, and hospital stay, as well as the presence of home oxygen therapy, were recorded.

### 2.5. Statistical Analysis

All analyses were performed in accordance with the intention-to-treat principle. The baseline characteristics and outcomes of the patients were expressed as mean (standard deviation), median (range: minimum–maximum), and frequency (percentage) for the comparison between the two groups. The binary data were analyzed using the chi-square test or Fisher’s exact test, whereas the continuous data were analyzed using Student’s *t*-test. The Kaplan–Meier curves were used to illustrate the event rates of the primary outcome, with the log-rank test utilized for analysis. The hazard ratios were calculated using Cox regression to compare the incidence rates of the primary outcome between the two groups. Statistical significance was set at a *p* value of <0.05. SPSS version 11.01 and JMP version 6.0 were used for statistical analysis.

## 3. Results

### 3.1. Study Participants

The flow diagram depicts the screening process for infants enrolled in the study, which entailed a random assignment to receive either TP tube feeding or NG tube feeding (Figure 1). Of the 100 infants initially recruited for the study, 51 did not meet the inclusion criteria, while 10 were excluded due to mortality before randomization. The remaining 39 infants were randomly assigned to either the TP or NG group, with 20 infants assigned to the TP group and 19 to the NG group.

### 3.2. Demographic and Baseline Characteristics

The demographic and clinical characteristics of the mothers and infants did not differ between the two groups (Table 1). The mean entry day of the study was Day 11.8 ± 3.0 from birth in the TP group and Day 10.1 ± 2.7 from birth in the NG group. However, no significant difference was observed.

### 3.3. Incidence of Adverse Events

The hazard ratio for primary adverse events, including BPD at 36 weeks, was significantly lower in the TP group than in the NG group (0.27, 95% CI 0.08–0.75; likelihood ratio test (*p* = 0.012), log rank test (*p* = 0.01)). The median time to event was 34 days (range: 24–85 days) with an incidence of 5/20 in the TP group. In contrast, the median time to event was 24 days (range: 13–70 days) with an incidence of 11/19 in the NG group (Figure 2). Table 2 shows the details of the primary outcomes observed in the study. At 36 weeks, three infants (15%) in the TP group and seven (37%) in the NG group have developed BPD. None of the infants in the TP group showed respiratory deterioration, whereas in the NG group, eight infants (42%) showed respiratory deterioration, and one infant died of respiratory failure on Day 10.2 ± 5.1. Only three infants (15%) in the TP group showed mild intestinal disorders on Day 18.3 ± 4.9, whereas one infant (5%) in the NG group showed a similar condition.

### 3.4. Incidence of Prematurity-Related Complications

Neither group presented with severe intestinal complications. Table 3 shows the incidence of prematurity-related complications. No differences were observed between the groups in relation to prematurity-related complications. However, the number of days of supplemental oxygen in the NG group was longer than that in the TP group.

### 3.5. Impacts of Tube Feedings on Growth and Respiratory Variables

No significant differences in the growth variables were observed between the two groups of infants, including body weight; head circumference at 4, 8, 12, and 16 weeks; and feeding intake (Table 4). The respiratory variables of the subgroup that were switched from NG tube feeding to TP tube feeding were found to be inferior to those of the other subgroups (Table 5). The day of switching between the two feeding methods due to adverse events occurred on Day 31.0 ± 6.1 after birth in the TP group and Day 21.2 ± 5.5 in the NG group (the difference was not statistically significant).

### 3.6. Impacts of Tube Feedings on Oxygenation Factors

Nine infants in the NG group showed significantly poorer oxygenation factors, including FiO_2_ on Days 7 and 14, FSp95 on Day 14, and RAS on Day 7, compared with infants in the TP group. Following the switch to TP feeding on Day 10.2 ± 5.2, the infants showed an immediate improvement in oxygenation factors (Figure 3).

## 4. Discussion

The findings of our study suggested that TP tube feeding is a safer alternative method of NG tube feeding for intubated ELBWIs. Our results revealed that TP tube feeding resulted in a reduction in adverse events, including intestinal disorders and respiratory deterioration, including BPD at 36 weeks postmenstrual age in ELBWIs.

A Cochrane review of nine eligible trials involving a total of 359 preterm infants revealed an absence of evidence indicating a beneficial effect of TP tube feeding in comparison with NG tube feeding among preterm infants. The meta-analysis also revealed that infants who underwent TP tube feeding showed a higher risk of gastrointestinal disturbance, including abdominal distention, gastric bleeding, diarrhea, and all-cause mortality. However, no statistically significant differences were observed in the incidence of other adverse events, including NEC, intestinal perforation, and aspiration pneumonia. Moreover, BPD was not reported in any of the trials [14].

However, it should be noted that the studies included in the review were not entirely free from bias. First, all trials were of a small scale and lacked a power or sample size calculation, which may have introduced a degree of bias. Second, the allocation was not concealed, and this may have affected the outcome independent of the intervention. Third, the largest study revealed differences in the baseline characteristics of the feeding groups, which may have influenced the results [22]. The exclusion of this study from the sensitivity analysis would result in a reduction in the relative risk of the incidence of gastrointestinal disturbance and mortality rates, from 1.5 to 1.4 and from 2.5 to 2.2, respectively. Fourthly, the validity of the data in the review was constrained by the absence of a comprehensive follow-up in the included trials. The included trials did not adhere to the intention-to-treat principle. In comparison with this systematic review, our study was not of a large scale. It was a pilot randomized controlled trial conducted at a single center. Furthermore, all cases included in the study underwent the allocated tube procedure and were followed up until the end of the study.

The objective of managing ELBWIs has undergone a significant transformation. Initially focused on survival, the current emphasis is on “intact survival” without complications. This shift was largely due to advancements in perinatal technology. Consequently, the mortality rate in ELBWIs, which represents a significant concern in the systematic review, has declined. However, the incidence of BPD has increased and now represents a significant challenge in perinatal care [23,24,25,26,27]. A nationwide cohort study using data from the affiliated hospitals of the Neonatal Research Network of Japan revealed a significant decline in the mortality rate among ELBWIs, from 19.0% in 2003 to 8.0% in 2016. In contrast, among the 17,126 survivors, 7792 (45.5%) infants have developed BPD, representing a significant increase from 41.4% in 2003 to 52.0% in 2016 [23]. Various treatments for BPD, including antenatal corticosteroids, surfactant, diuretics, inhaled bronchodilators, Phosphodiesterase-5 inhibitors, inhaled pulmonary vasodilators, endothelin-1 receptor antagonists, and macrolide antibiotics have been used in clinical settings but have not been effective in reducing the incidence of BPD [28]. In light of this, we undertook a reassessment of TP tube feeding as a potential strategy for BPD prevention.

GER is considered a major factor in tracheal silent aspiration in intubated preterm infants. In these infants, gastric contents can easily flow backward into the oral cavity through the lower esophageal sphincter, subsequently entering the trachea around the endotracheal tube [9]. The occurrence of a minimal amount of tube-mediated aspiration is considered a mechanism of deterioration in premature infants with BPD due to silent aspiration, which is a distinct phenomenon from acute aspiration syndrome that is caused by massive aspiration [8,10,29]. In the context of treating silent aspiration, it is important to control GER in order to reduce the incidence of apnea and the necessity for supplemental oxygen and to improve the appearance of chest X-rays [10]. In our study, we observed an immediate improvement in oxygenation factors in infants who were switched from NG tube feeding to TP tube feeding to control GER.

TP tube feeding has been used as a method of enteral feeding for over three decades. In view of the advances in perinatal technology and the enhanced quality of TP tube devices, several studies have reported the beneficial impact of TP tube feeding. Dryburgh documented the positive outcomes of TP tube feeding in 49 infants, including eight ELBWIs who exhibited good weight gain and were generally secure with this method [17]. Similarly, Malcolm et al. have observed that TP tube feeding, particularly when restricted to human milk, may effectively reduce the occurrence of apnea and bradycardia in preterm infants with suspected GER [30]. In a recent retrospective study, Srivatsa et al. investigated the impact of TP feed initiation on short-term oxygenation and manual oxygen blender titration among ELBWIs, with a sample size of 56 infants. The findings indicated that no significant differences were observed in any oxygenation measures during TP tube feeding vs. NG tube feeding among the 14 intubated infants. However, among the 42 nonintubated patients, significant improvements were observed in the SpO_2_/FiO_2_ ratios, frequency of manual oxygen titrations, number of hypoxemic episodes, and severity of hypoxemic episodes after TP tube placement. It was concluded that the transition from NG tube feeding to TP tube feeding was associated with acute improvement in oxygenation for nonintubated infants (n = 42) but not for intubated infants (n = 14) [31]. In contrast, Shimokaze et al. also evaluated the short-term effects of TP tube feeding on respiratory status in 33 preterm infants during mechanical ventilation and concluded that TP tube feeding during mechanical ventilation could prevent the deterioration of oxygenation without major complications at the stage of respiratory exacerbation in preterm infants [32]. The discrepancy in the efficacy of TP tube feeding on oxygenation in intubated preterm infants between the two studies may be attributed to several factors, including differences in data collection methods, ventilation modes (such as conventional ventilation and high-frequency ventilation), and the number of enrolled ventilated infants (14 vs. 33). Our findings align with the acute respiratory effects of TP tube feeding for respiratory exacerbation in preterm infants, as previously reported by Shimokaze and colleagues.

This study is limited in several ways. First, it is a single-center pilot study with a small sample size. Second, the data collected during the study period (2001–2003) may not be representative of current neonatal care practices and populations. However, there are few studies investigating the efficacy of early TP tube feeding in preventing adverse respiratory events in ELBWIs. Therefore, this study could contribute to strengthening the evidence base by providing data for a systematic review and meta-analysis. Furthermore, the long-term outcomes of included patients, such as neurodevelopmental and respiratory outcomes, will be incorporated in the future.

## 5. Conclusions

We concluded that early TP tube feeding may be a more effective alternative method of NG tube feeding. This is evidenced by a reduction in adverse respiratory events, including BPD, which may be attributed to the potential for silent aspiration in ELBWIs. However, due to the limited sample size in our pilot study, further confirmation of these findings is necessary. To this end, a large-scale multicenter randomized controlled trial is required to assess the clinical safety and efficacy of early TP tube feeding in ELBWIs.

## Figures and Tables

**Figure 1 biomedicines-12-02799-f001:**
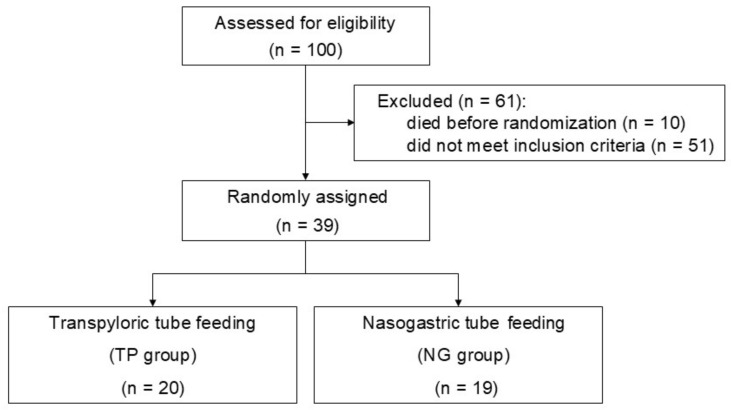
Flow diagram of the study participants.

**Figure 2 biomedicines-12-02799-f002:**
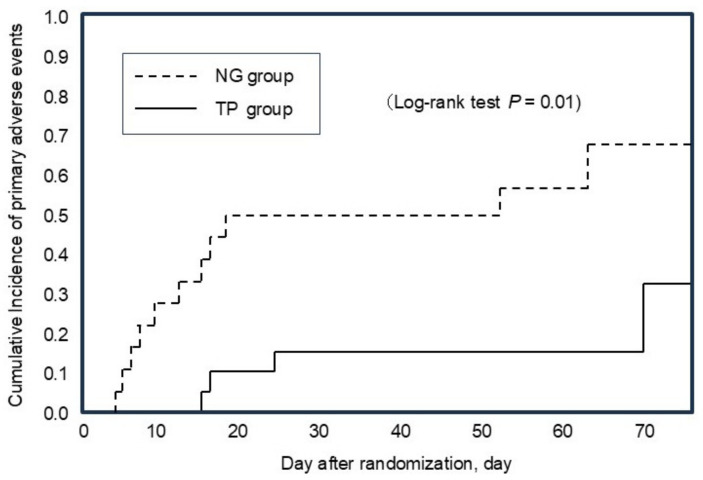
Kaplan–Meier estimates of the cumulative incidence of primary adverse events.

**Figure 3 biomedicines-12-02799-f003:**
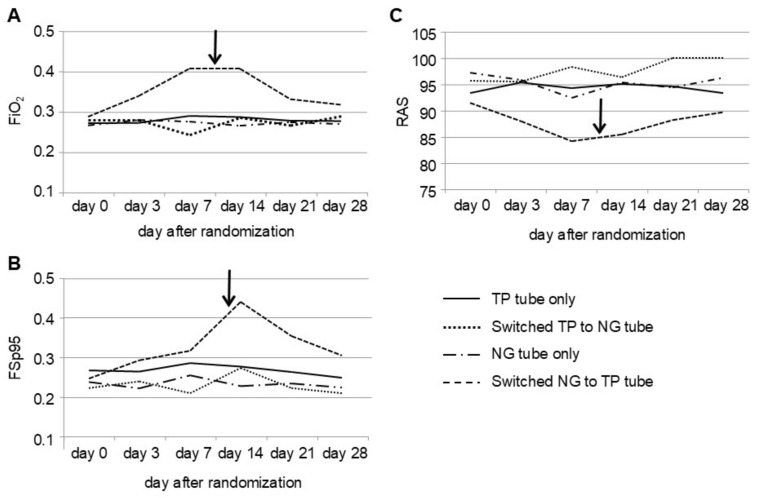
Course of oxygenation factors. (**A**), FiO_2_; (**B**), FSp95; and (**C**), RAS. The NG group showed significantly poorer oxygenation factors, including FiO_2_, FSp95, and RAS, compared with the TP group. However, after these infants were switched to TP feeding, they showed an immediate improvement in oxygenation factors. NG, nasogastric tube feeding; FSp95, FiO_2_ to keep the SpO_2_ at ≥95% at rest; RAS, room air saturation; TP, transpyloric. The arrow indicates the date on which the NG tube was replaced with the TP tube.

**Table 1 biomedicines-12-02799-t001:** Demographic and baseline characteristics.

	TP Group (n = 20)	NG Group (n = 19)	*p*
PROM, n (%)	9 (45)	5 (26)	0.23
Chorioamnionitis, n (%)	10 (50)	7 (37)	0.42
Prenatal steroid treatment, n (%)	14 (70)	12 (63)	0.66
Gestational age, mean (SD), wk	25.6 (1.4)	25.6 (1.2)	0.98
Birth weight, mean (SD), g	762 (157)	716 (126)	0.31
Birth head circumference, mean (SD), cm	23.2 (1.7)	22.6 (1.4)	0.78
Male, n (%)	7 (35)	8 (42)	0.66
Small for gestational age, n (%)	2 (10)	2 (11)	0.96
Multiplicity of birth, n (%)	5 (25)	3 (16)	0.49
Caesarean section, n (%)	16 (80)	13 (68)	0.42
Apgar score < 6 at 5min, n (%)	3 (15)	2 (11)	0.69

TP, transpyloric tube feeding; NG, nasogastric tube feeding; PROM, premature rupture of membranes; SD, standard deviation.

**Table 2 biomedicines-12-02799-t002:** Contents of primary outcome.

	TP Group (n = 20)	NG Group (n = 19)	*p*
Incidence of primary adverse events	5 (25)	11 (58)	0.05
BPD at 36 wk, n (%)	3 (15)	7 (37)	0.16
Adverse events			
Death, n (%)	0 (0)	1 (5) *	0.49
Respiratory deterioration, n (%)	0 (0)	8 (42) *	0.00
Intestinal disorder			
NEC stage 2≤, n (%)	0 (0)	0 (0)	1.00
NEC stage 1, n (%)	3 (15)	1 (5)	0.61

* Died of respiratory deterioration; TP, transpyloric tube feeding; NG, nasogastric tube feeding; BPD, bronchopulmonary dysplasia; NEC, necrotizing enterocolitis.

**Table 3 biomedicines-12-02799-t003:** Prematurity-related complications.

	TP Group (n = 20)	NG Group (n = 19)	*p*
Pneumonia, n (%)	2 (10)	2 (11)	0.96
Pulmonary hemorrhage, n (%)	1 (5)	2 (11)	0.96
Pneumothorax/PIE, n (%)	0 (0)	1 (5)	0.33
Times of surfactant administration, times median (min–max)	1 (1–4)	1 (1–5)	0.45
Days of supplemental oxygen, days median (min–max)	50.5 (26–344)	66 (20–476)	0.38
Days of mechanical ventilation, days median (min–max)	48 (19–145)	52 (20–101)	0.78
Home oxygen therapy, n (%)	3 (15)	4 (21)	0.64
IVH ≥ grade 3, n (%)	2 (10)	1 (5)	0.59

Two cases of IVH in the TP group were onset on day 0-1 (before randomization). PIE, pulmonary interstitial emphysema; IVH, intraventricular hemorrhage.

**Table 4 biomedicines-12-02799-t004:** Growth variables.

	TP Group (n = 20)	NG Group (n = 19)	*p*
Day of start to feeding (SD), day	2.0 (1.3)	1.7 (1.2)	0.51
Day of feeding			
at 50 mL/kg/day (SD), day	11.6 (3.0)	10.1 (2.8)	0.11
at 100 mL/kg/day (SD), day	15.4 (3.6)	17.6 (8.5)	0.30
Body weight			
at 4 wk (SD), g	799 (148)	763 (158)	0.48
at 8 wk (SD), g	1099 (289)	1057 (206)	0.60
at 12 wk (SD), g	1692 (478)	1649 (423)	0.77
at 16 wk (SD), g	2495 (599)	2212 (602)	0.23
Head circumference			
at 4 wk (SD), cm	24.3 (1.5)	24.5 (1.5)	0.67
at 8 wk (SD), cm	27.7 (2.3)	27.5 (1.7)	0.73
at 12 wk (SD), cm	29.9 (2.6)	30.8 (2.4)	0.33
at 16 wk (SD), cm	33.6 (2.0)	33.3 (1.9)	0.71
Rate of maximal weight loss (SD), %	13.4 (5.9)	15.0 (5.7)	0.41
Day of return to birth weight (SD), day	20.5 (9.5)	22.6 (10.1)	0.49

**Table 5 biomedicines-12-02799-t005:** Respiratory variables.

Allocated Groups	TP Tube (n = 20)		NG Tube (n = 19)	
Subgroups	TP Tube Only (n = 17)	Switch to NG Tube (n = 3)	NG Tube Only (n = 10)	Switch to TP Tube (n = 9)
Times of surfactant administration, times median (min–max)	1.0 (1–4)	1.0 (1–3)	1.0 (1–2)	3.0 (1–5)
Days of supplemental oxygen, days median (min–max)	48 (26–344)	84 (26–129)	44 (31–114)	97 (20–476)
Days of mechanical ventilation, days median (min–max)	43 (19–145)	50 (49–63)	35.5 (23–101)	58 (20–78)
Home oxygen therapy, n (%)	2 (12)	1 (33)	0 (0)	4 (44)

## Data Availability

The raw data supporting the conclusions of this article will be made available by the authors on request.

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
