# Peer review of "Early Transpyloric Tube Feeding in Preventing Adverse Respiratory Events in Extremely Low Birth Weight Infants"

_biomedicines, 2024, doi:10.3390/biomedicines12122799_

Round 1

Reviewer 1 Report

Comments and Suggestions for Authors

The study was devoted to clinical issues in neonatal care, focusing on improving respiratory outcomes in extremely low birth weight infants.  The study has a good design.

However, the main concerns with this study are the small sample size (39 newborns) and the study's time frame (2001-2003). However, the authors emphasize these limitations.

Some suggestions for improvements:

1. Concerns about its relevance to current neonatal care practices. Highlighting how perinatal technology advancements might influence the findings since then would strengthen the discussion.

2. Also, attention should be paid to the references. Only 4 references (15%) are from the last 10 years, the rest cover the period from 1986 to 2009. More new references should be added and the current state of the problem should be analyzed in the discussion. 

3. The study mentions stratification but does not provide a detailed analysis of potential confounding factors, such as antenatal steroid use or neonatal sepsis, which may affect respiratory outcomes.

4. Table 2 could be enhanced by including p-values to clarify the statistical significance of the findings.

Author Response

Comments 1: Concerns about its relevance to current neonatal care practices. Highlighting how perinatal technology advancements might influence the findings since then would strengthen the discussion.

Response 1: As the reviewer 1 noted, we have incorporated the latest developments in perinatal technology, particularly pharmacotherapy, into the discussion section. Despite these advancements, the rate of bronchopulmonary dysplasia remains unaltered.

“Various treatments for BPD, including antenatal corticosteroids, surfactant, diuretics, inhaled bronchodilators, Phosphodiesterase-5 inhibitors, inhaled pulmonary vasodi-lators, endothelin-1 receptor antagonists, and macrolide antibiotics have been used in clinical settings but have not been effective in reducing the incidence of BPD [Sakaria RP, Dhanireddy R. Pharmacotherapy in Bronchopulmonary Dysplasia: What Is the Evidence? Front Pediatr. 2022 Mar 9;10:820259.].”

Comments 2: Also, attention should be paid to the references. Only 4 references (15%) are from the last 10 years, the rest cover the period from 1986 to 2009. More new references should be added and the current state of the problem should be analyzed in the discussion.

Response 2: As per the feedback from Reviewer 1, we have incorporated the recently published references into the discussion section.

  • Watson J, McGuire W. Transpyloric versus gastric tube feeding for preterm infants. Cochrane Database Syst Rev. 2013 Feb 28;2013(2):CD003487.
  • Jensen EA, Munson DA, Zhang H, Blinman TA, Kirpalani H. Anti-gastroesophageal reflux surgery in infants with severe bronchopulmonary dysplasia. Pediatr Pulmonol. 2015 Jun;50(6):584-7.

Comments 3: The study mentions stratification but does not provide a detailed analysis of potential confounding factors, such as antenatal steroid use or neonatal sepsis, which may affect respiratory outcomes.

Response 3: At the time of randomization, patients were stratified according to the presence of intrauterine infection or inflammation, which has the potential to significantly impact respiratory outcomes. Due to the limited sample size of the pilot RCT, we did not analyze the data using stratification based on antenatal steroids or neonatal sepsis, as recommended by reviewer 1.

Comments 4: Table 2 could be enhanced by including p-values to clarify the statistical significance of the findings.

Response 4: As per the feedback from Reviewer 1, we have included p-values for each comparison.

Reviewer 2 Report

Comments and Suggestions for Authors

GENERAL

-          The main issue with this paper is that the data is from twenty years ago. The authors need to make a strong argument to explain why they published old research.

ABSTRACT

-          There was no background in the abstract. It was only the aim.

-          I suggest adding 1-2 sentences about the subjects and replacing the sentence regarding ethical permission.

KEYWORDS

-          I suggest adding “adverse events.”

INTRODUCTION

-          The introduction did not include data regarding previous studies, especially the systematic review regarding this technique. However, the discussion did provide an explanation. If there was at least one systematic review about a similar topic, then the authors should explain why they redone the trial.

-          Please add some more recent publications.

METHODS

-          Line 74: Please elaborate more about the stratification of the subjects

-          Line 79: What was a “sealed, opaque envelope”? Why and how was it used in this trial?

-          The trial's sample size is not calculated. The authors mentioned the study's power in the discussion, but I did not see the data in the methods section.

-          Point 2.3 Intervention: The authors only mentioned about transpyloric tube but there was no explanation regarding the nasogastric tube

-          Line 129: The recorded data was ended at 28 days. Was it true?

RESULTS

-          See comments on tables and figures

DISCUSSION

-          Line 228: How do the authors avoid bias in this study?

-          Line 238; Sufficient statistical powers was not supported by any data

-          The authors should add the negative aspects of the transpyloric tube technique to make it more balanced

REFERENCES

-          Reference number 1 was incorrect

-          All references but the last five were ancient

TABLES

-          Table 1: Please add an explanation of the abbreviation

-          Table 2: Please add the p-value

FIGURES

-          Figure 1: Box “excluded” should not mention “did not meet inclusion criteria”.

Author Response

Comments 1: The main issue with this paper is that the data is from twenty years ago. The authors need to make a strong argument to explain why they published old research.

Response 1: This manuscript was not submitted for publication following the presentation at the conference. After a period of 20 years, we believe that this manuscript should be published not only for the patients included in the study but also for the benefit of the wider field of neonatal medicine. We have added some sentences to explain why this study should be published.

“Second, the data collected during the study period (2001–2003) may not be representative of current neonatal care practices and populations. However, there are few studies investigating the efficacy of early TP tube feeding in preventing adverse respiratory events in ELBWIs. Therefore, this study could contribute to strengthening the evidence base by providing data for a systematic review and meta-analysis. Furthermore, the long-term outcomes of included patients, such as neurodevelopmental and respiratory outcomes, will be incorporated in the future.”

Comments 2: There was no background in the abstract. It was only the aim.

Response 2: One sentence has been included in the background of the abstract.

“It has been demonstrated that aspiration during endotracheal intubation in preterm infants with gastroesophageal reflux is a contributing factor in the worsening of lung diseases and the development of bronchopulmonary dysplasia (BPD).”

Comments 3: I suggest adding 1-2 sentences about the subjects and replacing the sentence regarding ethical permission.

Response 3: The information regarding the subjects has been added, and the sentence regarding registration has been deleted.

“The study population consisted of 39 ELBWIs with mechanical ventilation and an enteral feeding volume of 50 ml/kg/day, which were randomly assigned to different groups based on the method of tube feeding.”

Comments 4: I suggest adding “adverse events.”

Response 4: In accordance with the recommendation from Reviewer 2, the term "adverse events" has been included as a keyword.

Comments 5: The introduction did not include data regarding previous studies, especially the systematic review regarding this technique. However, the discussion did provide an explanation. If there was at least one systematic review about a similar topic, then the authors should explain why they redone the trial.

Response 5: We have included a Cochrane review published in 2013 in the introduction section for reference.

Comments 6: Please add some more recent publications.

Response 6: We have recently updated our publications.

Comments 7: Line 74: Please elaborate more about the stratification of the subjects

Response 7: The sentences have been revised for greater clarity and ease of comprehension.

“The patients were stratified by the presence of intrauterine infection/inflammation (high neonatal serum IgM levels, chorioamnionitis, and a negative gastric stable microbubble test [16]) and gestational age (three blocks: 22–23, 24–26, and 27 weeks).”

Comments 8: Line 79: What was a “sealed, opaque envelope”? Why and how was it used in this trial?

Response 8: A sealed, opaque envelope was utilized for the randomization process. The sentences have been modified for enhanced readability.

Comments 9: The trial's sample size is not calculated. The authors mentioned the study's power in the discussion, but I did not see the data in the methods section.

Response 9: The sentences pertaining to the study's power in the discussion section have been removed. This is in accordance with the recommendation of Reviewer 2, who highlighted the absence of a sample size calculation in the pilot study.

Comments 10: Point 2.3 Intervention: The authors only mentioned about transpyloric tube but there was no explanation regarding the nasogastric tube

Response 10: The following sentences have been incorporated into the text,

“The NG tube was also meticulously positioned within the nasal cavity and advanced into the gastric region, subsequent to the determination of the requisite length of tube necessary to achieve gastric access in the infant.”

Comments 11: Line 129: The recorded data was ended at 28 days. Was it true?

Response 11: The data recorded on Days 0, 3, 7, 14, 21, and 28 were used in Figure 3. However, the data were recorded on a daily basis until the postmenstrual age of 36 weeks. Therefore, we have amended the sentence.

“These data were recorded on a daily basis until the postmenstrual age of 36 weeks.”

Comments 12: See comments on tables and figures

Response 12: As Reviewer 2 correctly observed, we have implemented the suggested changes.

Comments 13: Line 228: How do the authors avoid bias in this study?

Response 13: As previously stated, “It was a pilot randomized controlled trial conducted at a single center. Furthermore, all cases included in the study underwent the allocated tube procedure and were followed up until the end of the study”. It is inevitable that a pilot study with a limited sample size will be susceptible to certain biases.

Comments 14: Line 238; Sufficient statistical powers was not supported by any data

Response 14: As reviewer 2 correctly noted, the pilot study lacked sufficient statistical power to support the data presented. Consequently, we have deleted the aforementioned sentence.

Comments 15: The authors should add the negative aspects of the transpyloric tube technique to make it more balanced

Response 15: As per the feedback from Reviewer 2, the sentence in the conclusion section has been amended.

“We concluded that early TP tube feeding may be a more effective alternative method of NG tube feeding. This is evidenced by a reduction in adverse respiratory events, including BPD, which may be attributed to the potential for silent aspiration in ELBWIs. However, due to the limited sample size in our pilot study, further confirmation of these findings is necessary. To this end, a large-scale multicenter randomized controlled trial is required to assess the clinical safety and efficacy of early TP tube feeding in ELBWIs.”

Comments 16: Reference number 1 was incorrect

Response 16: We have deleted reference number 1.

Comments 17: All references but the last five were ancient

Response 17: As previously stated in the limitations section, this RCT is an older pilot study. Therefore, we have included older articles in the introduction section. We have added a couple of recent articles in the discussion section in response to the reviewer 2's comments.

Comments 18: Table 1: Please add an explanation of the abbreviation

Response 18: We have added an explanation of the abbreviation at the bottom of Table 1.

Comments 19: Table 2: Please add the p-value

Response 19: We have included the p-value in Table 2.

Comments 20: Figure 1: Box “excluded” should not mention “did not meet inclusion criteria”.

Response 20: Our flow diagram in Figure 1 was created in accordance with the Consolidated Standards of Reporting Trials (CONSORT) 2010 guidelines. In the CONSORT 2010 diagram, the "excluded" box includes those who did not meet the inclusion criteria (n=).

Reviewer 3 Report

Comments and Suggestions for Authors

1.       how do you propose to optimize the timing and criteria for initiating TP tube feeding to maximize its benefits?

2.       Due to small sample size and the stratification how do you ensure sufficient statistical power to draw robust conclusions?

3.       How do you interpret the significant reduction in BPD and respiratory deterioration in the TP group despite the small sample size and overlapping confidence intervals for some outcomes?

4.       could you elaborate on improvement in oxygenation were sustained long-term

5.       Srivatsa et al., appear to show inconsistent outcomes between intubated and non-intubated infants with TP tube feeding. How do you reconcile these conflicting findings?

6.       how do you propose to address potential confounding factors like variations in feeding protocols and patient characteristics? Is it be generalized?

Author Response

Comments 1: How do you propose to optimize the timing and criteria for initiating TP tube feeding to maximize its benefits?

Response 1: While this is a pilot study with a limited sample size, we believe that our inclusion criteria (enteral feeding volume of 50 ml/kg/day in ELBWIs with mechanical ventilation) represents the optimal timing for initiating TP tube feeding to date.

Comments 2: Due to small sample size and the stratification how do you ensure sufficient statistical power to draw robust conclusions?

Response 2: As reviewer 3 correctly noted, the pilot study's small sample size prevented us from ensuring sufficient statistical power. To draw robust conclusions, a further large-scale RCT will be required.

Comments 3: How do you interpret the significant reduction in BPD and respiratory deterioration in the TP group despite the small sample size and overlapping confidence intervals for some outcomes?

Response 3: As reviewer 3 correctly observed, the efficacy of TP tube feeding cannot be strongly concluded due to the limited sample size. Consequently, we have included additional commentary in the conclusion section to temper its impact on respiratory status.

“However, due to the limited sample size in our pilot study, further confirmation of these findings is necessary. To this end, a large-scale multicenter randomized controlled trial is required to assess the clinical safety and efficacy of early TP tube feeding in ELBWIs.”

Comments 4: Could you elaborate on improvement in oxygenation were sustained long-term

Response 4: In this study, we didn’t follow their long-term outcome. As you suggested, we are going to collect data on the long-term respiratory outcomes.

This study did not include a long-term outcome assessment. As you suggested, we will be collecting data on long-term respiratory outcomes.

Comments 5: Srivatsa et al., appear to show inconsistent outcomes between intubated and non-intubated infants with TP tube feeding. How do you reconcile these conflicting findings?

Response 5: We have incorporated an additional paper by Shimokaze and addressed this topic in the discussion section.

Comments 6: How do you propose to address potential confounding factors like variations in feeding protocols and patient characteristics? Is it be generalized?

Response 6: To mitigate the impact of potential confounding factors, we conducted a randomized controlled trial (RCT). However, this is a pilot study. As reviewer 3 noted, a larger RCT will be necessary to generalize the findings.

Reviewer 4 Report

Comments and Suggestions for Authors

The authors conducted a study aimed at comparing the safety and efficacy of early transpyloric (TP) tube feeding with that of nasogastric (NG) tube feeding in relation to bronchopulmonary dysplasia (BPD), recruiting 39 ELBWIs (extremely low birth weight infants). The authors demonstrated that early TP tube feeding may be a safer alternative to NG tube feeding for intubated ELBWIs and has been shown to reduce the frequency of adverse respiratory events.  

This is an interesting manuscript, considering the incidence of bronchopulmonary dysplasia in preterm neonates and the severe impact on the quality of life of these patients.  

Below are my comments:  

- In the abstract, specify the acronym ELBWIs in full upon its first mention.  

- In the reference list, correct the formatting error (Reference 1 is misplaced as Reference 2, and so on).  

- Line 91 – Italicize the names of the lactobacilli strains used.  

- The statistical analysis and ethical committee approval are appropriate; however, in Line 301, if possible, specify the protocol number for the ethical approval.  

- Line 156 – Are there any available data on maternal vaginal colonization (positive/negative for Streptococcus agalactiae)? Including this information in the table or main text would be valuable.  

Author Response

Comments 1: In the abstract, specify the acronym ELBWIs in full upon its first mention. 

The acronym ELBWIs has been defined in the abstract.

Response 1: In the reference list, correct the formatting error (Reference 1 is misplaced as Reference 2, and so on).

We have conducted a thorough review and implemented the necessary corrections.

Comments 2: Line 91 – Italicize the names of the lactobacilli strains used. 

Response 2: In accordance with the recommendation of Reviewer 4, the names of the lactobacilli strains have been changed to italic.

Comments 3: The statistical analysis and ethical committee approval are appropriate; however, in Line 301, if possible, specify the protocol number for the ethical approval. 

Response 3: In accordance with the recommendation from Reviewer 4, we have included the trial ID (UMIN000001728) in the submission.

Comments 4: Line 156 – Are there any available data on maternal vaginal colonization (positive/negative for Streptococcus agalactiae)? Including this information in the table or main text would be valuable. 

Response 4: We regret to inform you that we do not have the necessary data on the specific strains of maternal vaginal colonization.

Round 2

Reviewer 2 Report

Comments and Suggestions for Authors

Thank you for modifying the paper. 

Reviewer 3 Report

Comments and Suggestions for Authors

They were much improved. Almost all of my points are taken care of.